# Doxycycline Decreases Atherosclerotic Lesions in the Aorta of *ApoE*^-⁄-^ and Ovariectomized Mice with Correlation to Reduced MMP-2 Activity

**DOI:** 10.3390/ijms23052532

**Published:** 2022-02-25

**Authors:** Keuri E. Rodrigues, Aline Azevedo, Pricila R. Gonçalves, Maria H. B. Pontes, Gustavo M. Alves, Ruan R. Oliveira, Cristine B. Amarante, João P. M. Issa, Raquel F. Gerlach, Alejandro F. Prado

**Affiliations:** 1Institute of Biological Sciences, Federal University of Pará, Cardiovascular System Pharmacology and Toxicology Laboratory, Belém 66075-110, PA, Brazil; keuri13@yahoo.com.br (K.E.R.); pri.pharma.12@gmail.com (P.R.G.); helenapontesmb@gmail.com (M.H.B.P.); gustavo.g.ma@hotmail.com (G.M.A.); oliveiraruanfr@gmail.com (R.R.O.); 2Department of Biomechanics, Medicine and Locomotor Apparatus Rehabilitation, Faculty of Medicine of Ribeirão Preto, University of Sao Paulo, Ribeirão Preto 14049-900, SP, Brazil; aline_azevedo79@hotmail.com; 3Coordination of Earth Sciences and Ecology, Museu Paraense Emílio Goeldi, Belem 66077-830, PA, Brazil; cbamarante@museu-goeldi.br; 4Department of Basic and Oral Biology, School of Dentistry of Ribeirao Preto, University of Sao Paulo (FORP/USP), Ribeirao Preto 14040-904, SP, Brazil; jpmissa@forp.usp.br (J.P.M.I.); rfgerlach@forp.usp.br (R.F.G.)

**Keywords:** atherosclerosis, extracellular matrix, MMP inhibitor

## Abstract

Atherogenic events promote changes in vessel walls, with alteration of the redox state, and increased activity of matrix metalloproteinases (MMPs). Thus, this study aims to evaluate aortic remodeling, MMP activity, and reactive oxygen species (ROS) levels after treatment with doxycycline in *ApoE*^-⁄-^ and ovariectomized mice (OVX). Female *ApoE*^-⁄-^-knockout mice (5 weeks) were submitted to ovariectomy surgery to induce experimental menopause. They then received chow enriched with 1% cholesterol to induce hypercholesterolemia. The animals were divided into two experimental groups: *ApoE*^-⁄-^/OVX vehicle and *ApoE*^-⁄-^/OVX doxycycline (30 mg/kg) administered by gavage once a day for 28 days (15th to the 18th week of life). Blood samples were collected to measure total cholesterol and fractions. The aorta was used for morphometry and to measure the activity and expression of MMP-2 and ROS levels. The *ApoE*^-⁄-^/OVX doxycycline group showed no change in total and fraction cholesterol levels. However, there was a reduction in ROS levels, MMP-2 expression, and activity that correlated with a decrease in atherosclerotic lesions relative to the *ApoE*^-⁄-^/OVX vehicle (*p* > 0.05). Therefore, we conclude that doxycycline in *ApoE*^-⁄-^/OVX animals promotes a reduction in atherosclerotic lesions by reducing ROS and MMP-2 activity and expression.

## 1. Introduction

Cardiovascular diseases kill 17.9 million people a year worldwide [1], and atherosclerosis is the underlying cause of 45% of these deaths [2]. Atherosclerosis is a chronic inflammatory disease characterized by the accumulation of lipids in the artery walls with extracellular matrix deposition forming an atheromatous plaque, thus leading to stenosis and reduced blood flow [3,4,5].

The etiology of atherosclerosis is multifactorial [6,7]. The contemporary lifestyle often consists of an inadequate diet, sedentary lifestyle, smoking, and drinking, which induces changes in the lipid profile. This lipid dysregulation is a leading risk factor for the onset of atherogenesis [8,9]. Age is also indicated in the etiology of atherosclerosis because there is a decrease in the production of hormones such as estrogen with menopause, thus leading to an acceleration in oxidative and inflammatory processes and a reduction in the stimulation of nitric oxide and alteration in the lipid profile. This lipid dysregulation promotes the formation of atheromatous plaques and vascular lesions [10,11,12].

Genetic causes can lead to changes in apolipoproteins E (*ApoE*) [13,14]. *ApoE*^-⁄-^ animals have high levels of triglycerides and total cholesterol [15]. *ApoE*-knockout mice are more likely to develop atherosclerosis due to an accumulation of unprocessed lipids with difficulties in the endocytosis of IDL triglycerides. These mice have lower production of HDL [16,17,18]. *ApoE*^-⁄-^/OVX animals have increased plasma triglyceride levels associated with insulin dysregulation even on a lipid-free diet [19]. *ApoE*^-⁄-^-knockout animals that received a cholesterol-enriched diet showed a correlation between a lack of *ApoE* and increased fat intake, thus promoting the development of atherosclerosis [20].

The activity and expression of MMPs are also altered in atherosclerosis. Levels of MMP-2, MMP-3, MMP-8, MMP-9, and MMP-12 are elevated and participate in vascular smooth muscle cell migration, neointimal formation, and atheromatous plaque growth [20,21,22,23,24,25,26,27,28]. The transcriptional production of metalloproteinases is regulated by a range of pro-inflammatory cytokines, growth factors, hormones, cell–cell interactions, and cell-extracellular matrix (ECM) interactions. In addition, the post-translational activity of MMPs is regulated by tissue inhibitors of MMPs (TIMPs) and reactive oxygen species (ROS) [29].

Animals that were *ApoE*^-⁄-^ and *MMP-2*^-⁄-^ [23], *MMP-8*^-⁄-^ [20], *MMP-9*^-⁄-^, and *MMP- 12*^-⁄-^ [25] showed a decrease in atherosclerotic lesions. In this sense, the pharmacological inhibition of MMPs may have a beneficial effect on the treatment of atherosclerosis. Doxycycline is a tetracycline antibiotic that inhibits metalloproteases at sub-antimicrobial doses [30]. In an animal model of renovascular hypertension, doxycycline decreased blood pressure, cardiovascular remodeling, and MMP activity [31,32,33]. Doxycycline can also effectively decrease ROS [34,35] and increase NO bioavailability [35]. Furthermore, it is the only widely available MMP inhibitor in the clinic [30,36].

Thus, treatment with doxycycline in *ApoE*^-⁄-^/OVX animals can help prevent vascular remodeling and reduce atherosclerotic lesions triggered by hypercholesterolemia and menopause associated with increased ROS and MMP-2 activity [37]. Therefore, this study hypothesizes that doxycycline reduces atherosclerotic lesions in the aorta of *ApoE*^-⁄-^ ovariectomized mice (OVX) by inhibiting MMPs and reducing ROS. Thus, this study aims to evaluate aortic remodeling, MMP activity, and ROS levels after treatment with doxycycline in *ApoE*^-⁄-^/OVX mice.

## 2. Results

Blood and aorta samples obtained from wild C57BL6 experimental groups (Vehicle and Doxycycline) showed normal levels within what is recommended for total cholesterol, LDL, and HDL (data not shown). Therefore, we used the samples to establish a baseline for the other parameters evaluated: Morphometry, gelatinolytic activity, MMP-2 expression, and ROS levels (data not shown) (*p* < 0.05 vs. wild-type C57BL6). However, this research seeks to understand if treatment with doxycycline in an already-established model of atherosclerosis will have an antiatherogenic effect by modulating the activity and expression of MMP-2 and ROS. Thus, the results focus on the *ApoE*^-⁄-^/OVX (Vehicle and Doxycycline) groups.

### 2.1. Doxycycline Treatment Does Not Alter the Lipid Profile in ApoE^-⁄-^/OVX Mice

The plasma obtained from the experimental groups showed that the *ApoE*^-⁄-^/OVX animals had high levels of total and LDL cholesterol and low levels of HDL cholesterol. These metrics are associated with a hypercholesterolemic diet and genetic deletion of *ApoE*. Doxycycline treatment did not change the levels of total cholesterol and fractions relative to the *ApoE*^-⁄-^/OVX vehicle group (*p* > 0.05). These values are described in Table 1.

### 2.2. Doxycycline Reduced the Size of Atherosclerotic Lesions in ApoE^-⁄-^/OVX Animals

The hypercholesterolemic diet led to the appearance of atherosclerotic lesions in the thoracic aorta of *ApoE*^-⁄-^/OVX animals (Figure 1A–D). Doxycycline treatment decreased the area of atherosclerotic lesions in the *ApoE*^-⁄-^/OVX group relative to the vehicle group (*p* > 0.05, Figure 1A,D). No hypertrophy of the tunica media was observed as evidenced by the quantification of the cross-sectional area and the ratio of tunica media and lumen diameter between the two experimental groups (*p* > 0.05, Figure 1A–C).

### 2.3. Doxycycline Decreases Gelatinolytic Activity and MMP-2 Expression in ApoE^-⁄-^/OVX In Situ Animals

High gelatinolytic activity associated with increased MMP-2 was observed in *ApoE*^-⁄-^/OVX animals as measured by fluorescence intensity (Figure 2A,B and Figure 3A,B). The *ApoE*^-⁄-^/OVX doxycycline group had reduced tissue gelatinolytic activity (*p* < 0.05, Figure 2A,B) and reduced MMP-2 labeling in vascular tissue relative to the *ApoE*^-⁄-^/OVX vehicle group (*p* < 0.05, Figure 3A,B).

### 2.4. Doxycycline Treatment Decreases ROS in ApoE^-⁄-^/OVX Animals

*ApoE*^-⁄-^/OVX animals had higher levels of ROS in the aortas: There was high fluorescence intensity emitted by the DHE probe (*p* < 0.05, Figure 4A,B). The *ApoE*^-⁄-^/OVX doxycycline group showed decreased ROS levels in the aortas vs. the *ApoE*^-⁄-^/OVX vehicle (*p* < 0.05, Figure 4A,B).

The results described above showed that doxycycline has an antiatherogenic effect associated with a decrease in gelatinolytic activity, MMP-2 expression, and ROS levels in the aortas of *ApoE*^-⁄-^/OVX animals. The results are supported by the existence of positive correlations between increased gelatinolytic activity and ROS levels (r^2^ = 0.72, *p* = 0.0004, Figure 5A), as well as by the increase in MMP-2 expression and ROS levels (r^2^ = 0.65, *p* = 0.0014, Figure 5B) in the aortas of *ApoE*^-⁄-^/OVX animals, suggesting that MMP-2 is related to the increase in ROS in these animals. Additionally, we found positive correlations between increased atherosclerotic lesions associated with increased gelatinolytic activity (r^2^ = 0.33, *p* = 0.03, Figure 5C), MMP-2 expression (r^2^ = 0.41, *p* = 0.02, Figure 5D), and increased ROS (r^2^ = 0.33, *p* = 0.04, Figure 5E), confirming that the inhibition of MMP-2 by doxycycline promotes improvement in atherosclerotic lesions in the aorta of *ApoE*^-⁄-^/OVX mice.

## 3. Discussion

Our results show for the first time that treatment with doxycycline in *ApoE*^-⁄-^/OVX animals reduced atherosclerotic lesions associated with MMP-2 activity and expression and reduced ROS. However, such treatment is not associated with improved lipid profiles.

We found an increase in total and LDL cholesterol levels but low HDL levels in *ApoE*^-⁄-^/OVX animals. Our findings corroborate studies using *ApoE*^-⁄-^ animals and the OVX model. *ApoE*-knockout mice were more likely to develop atherosclerosis due to the accumulation of unprocessed lipids, difficulty in endocytosis of IDL triglycerides, and lower production of HDL [15,17,38,39]. The OVX model induces menopause with rapid and uniform estrogen deficiency, thus resulting in higher levels of triglycerides [40]. Several studies have shown that hormonal interruption interferes with the lipid profile, thus leading to oxidative and inflammatory processes that induce lesions in the vasculature and the development of atherosclerosis [10,18,41].

Here, we showed that the treatment of *ApoE*^-⁄-^/OVX animals with doxycycline reduced the gelatinolytic activity in the aortas. The MMP activity was evaluated by the in situ zymography, a technique described by Galis et al. in 1994 to show the MMP activity in atheroma plaques from humans with atherosclerosis [42]. The great advantage of this technique is the possibility of directly determining the net enzymatic activity in the intact tissue. Furthermore, it is a powerful tool associated with immunohistochemistry, as antibodies hardly differentiate between active and inactive MMPs in tissues [43,44]. Our results corroborate studies that show an increase in gelatinolytic activity in the aortas of atherosclerotic animals and humans [42,43,45,46,47]. Classically, the activity of MMPs can also be evaluated by the gel zymography technique described in 1980 [48], which is still widely used [49,50,51,52,53,54,55]. The increase in MMP-2 activity by gel zymography is shown in different models of atherosclerosis, which demonstrates that the rise in the activity of this gelatinase results in the progression of atherosclerotic lesions [23,55,56].

The decrease in gelatinolytic activity in the aorta of *ApoE*^-⁄-^/OVX animals by doxycycline was accompanied by a reduction in MMP-2 expression. Our results are similar to previous studies that show that atherogenesis increases the expression of MMP-2 in vascular tissue, which is related to the progression of atherosclerotic lesions [23,42,43,46,47,57].

MMPs can cleave several components of the ECM, including collagen and elastin [58] and cleavage of arterial wall components leads to migration of inflammatory cells. However, inflammatory cell infiltrates in the atherosclerotic lesion result in a significant increase in MMP activity and expression [49,59]. The profile of MMPs expressed within an atherosclerotic lesion has consequences on the composition of the ECM. MMP-2, MMP-3, MMP-8, MMP-9, and MMP-12 were associated with the cell migration process in vascular smooth muscle, thus favoring the growth of the atheromatous plaque [20,21,22,23,24,25,26,27,28]. Other studies have shown that the involvement of MMPs in atherosclerosis leads to a reduction in atherosclerotic lesions when using double knockout animals for *ApoE*^-⁄-^ and *MMP-2*^-⁄-^ [23], *MMP-8*^-⁄-^ [20], and *MMP-9*^-⁄-^ and *MMP-12*^-⁄-^ [25].

Here, we investigated the effect of MMP inhibition via doxycycline, which, despite not having a lipid-lowering effect, reduced atherosclerotic lesions associated with decreased MMP-2 activity and expression. The antiatherogenic effects of doxycycline observed in our study were similar to previous studies that used doxycycline [60,61,62].

*ApoE*-knockout females treated with 1.5 mg/kg doxycycline did not show changes in blood cholesterol and triglycerides, but there was decreased gelatinolytic activity and fewer atherosclerotic lesions in the aorta [62]. *ApoE*^-⁄-^ females also treated with 1.5 mg/kg doxycycline showed anti-inflammatory activity as indicated by decreased levels of MCP-I, IL-6, IL-12, or serum amyloid A [60,61]. The intraperitoneal administration of 1.5 µg/kg doxycycline in *ApoE*^-⁄-^ animals fed a high-fat diet also reduced atherosclerotic lesions in the aorta in these animals by decreasing pro-inflammatory cytokines and MMPs activity [60,61].

On the other hand, a high doxycycline dose does not increase the antiatherogenic effect. *ApoE*^-⁄-^ animals with aortic valve calcification treated with 100 mg/kg of doxycycline had no lipid-lowering activity and did not affect the gelatinolytic activity of MMPs. It did not impact the expression of MMP-9 and MMP-12 and did not improve aortic blood flow in this model [63]. The antiatherogenic effect of doxycycline also seems to depend on the severity of the model. In *LDL*^-⁄-^ animals that received angiotensin, an animal model of atherosclerosis and abdominal aortic aneurysms, doxycycline was used at 30 mg/kg and did not reduce total cholesterol levels and fractions and atherosclerotic lesions. However, there were fewer abdominal aortic aneurysms [64].

The effect of different doses (3, 10, and 30 mg/kg) of doxycycline in hypertensive rat kidneys was evaluated. All assessed doses effectively decreased blood pressure, but only 30 mg/kg reduced morphological changes and endothelial dysfunction in the aorta [32].

These studies with doxycycline demonstrate that the drug acts in different ways and has a dose-dependent effect, correlated with the route of administration, the dose administered, and the severity of the vascular dysfunction model used. However, in most studies, doxycycline had an antiatherogenic effect and reduced atherosclerotic lesions, which supports our findings that demonstrate an antiatherogenic impact [60,61,62].

In our study, the reduction of MMP-2 activity and expression by doxycycline leads to a decrease in ROS [34,35]. However, doxycycline has also been shown to have a direct antioxidant effect against H_2_O_2_-induced damage in cardiomyocytes and reduced oxidative stress by 5 mg/kg in an isoproterenol model of heart failure in rodents [65]. It is known that ROS can modulate the activity and expression of MMP-2 [66,67,68], and the use of antioxidants strengthens this evidence [69,70]. On the other hand, it was demonstrated that MMP-2 can activate pro-oxidant pathways through the cleavage of pro-HB-EGF. When exogenous MMP-2 (recombinant) was infused into aortas, an evident rise in ROS was observed, indicating for the first time that MMP-2 does induce ROS increases, instead of being only activated by ROS as usually thought. Furthermore, the authors showed a correlation between increased vascular gelatinolytic activity and ROS levels, which are decreased by MMP inhibitors such as doxycycline and phenanthroline, thus suggesting that these inhibitors have antioxidant action by inhibiting MMP-2 [71].

Our results also showed a positive correlation between increased ROS levels, gelatinolytic activity, and MMP-2 expression in the aortas of *ApoE*^-⁄-^/OVX mice, suggesting that MMP-2 plays a role in ROS production. Supporting these findings, the treatment of atherosclerotic rabbits with carvedilol also decreased the expression and activity of metalloproteinases-2 and 9 in the abdominal aorta, resulting in a reduction in ROS [57]. In fact, ROS were shown as crucial elements in modulating matrix degradation via MMP-2 activation in areas of oxidative stress, contributing to the instability of atherosclerotic plaques [72].

We also found a positive correlation between the increased area of atherosclerotic lesions with higher gelatinolytic activity, MMP-2 expression, and ROS, showing that treatment with doxycycline in *ApoE*^-⁄-^/OVX animals has an antiatherogenic effect associated with the reduction of these factors. Similar results were observed in *ApoE*^-⁄-^/*MMP-2*^-⁄-^ mice where *MMP-2* gene deletion resulted in a significant reduction of atherosclerotic lesions in the sinus and aortic arch of the animals, demonstrating the role of MMP-2 in the development of atherosclerotic lesions [23]. Other models of atherosclerosis, such as the *LDLr*^-⁄-^*Apob100/100* mice, a model that resembles atherosclerosis in humans, also showed increased expression and activity of MMP-2 and 9 associated with the progression of atherosclerotic lesions [47]. In human aortas, increased activity and expression of MMP-2 in fatty streaks and atherosclerotic plaques has been demonstrated, which has been associated with the formation and progression of atherosclerotic lesions, which can result in fatal events. Studies with patients with atherosclerosis also show MMP-2 as a marker of injury and mortality [22,27,73,74], which reinforces the effects observed by doxycycline in this study as promising for further studies.

Our research corroborates previous findings in which doxycycline had an atherogenic effect, reducing the lesion area associated with a decrease in the MMP activity, MMP-2 expression, and ROS. There were no lipid-lowering effects. More studies are needed to establish adequate conditions for the clinical use of doxycycline.

## 4. Material and Methods

### 4.1. Animals

Female C57BL/6J *ApoE*^-⁄-^ knockout mice (four-week-old) were provided by the Animal Facility of the Faculty of Dentistry of Ribeirão Preto (FORP-USP). The animals were kept in individual cages in a temperature-controlled environment at 22 °C and 12 h light/dark cycle with water provided ad libitum. The project was submitted and approved by the Animal Research Ethics Committee of the Faculty of Dentistry, University of São Paulo—FORP/USP, n° 2014.1.1090.58.0.

### 4.2. Experimental Design

The animals were divided into two experimental groups: *ApoE*^-⁄-^/OVX vehicle and *ApoE*^-⁄-^/OVX doxycycline. In the 5th week of life, the animals underwent ovariectomy to induce experimental menopause. In that procedure, the animals were anesthetized with xylazine (10 mg/kg) and ketamine (90 mg/kg). They then underwent trichotomy and local antisepsis, before surgical access and tissue divulsion and evisceration of the ovaries followed by antibiotic prophylaxis for 48 h [75]. For the induction of atherosclerosis, a hypercholesterolemic diet was instituted from the 7th week of life until the end of the experimental protocol. Treatments with water and doxycycline 30 mg/kg were administered by gavage once a day and lasted for 28 days from the 15th to the 18th week of life [32]. Figure 6 represents the workflow of the experimental design of the study.

### 4.3. Obtaining Blood and Aorta Samples

The animals were anesthetized (1% isoflurane) and euthanized by decapitation after the 18th week of life. Blood was collected in tubes containing 4% EDTA and centrifuged at 1200× *g* for 15 min at 4 °C to separate the plasma for the subsequent measurement of total cholesterol and fractions using a commercial kit (Labtest, Lagoa Santa, Mg, Brazil). The thoracic region was opened, and the thoracic aorta was dissected and divided into two parts: One for morphometric analysis and the other for measuring ROS, activity, and MMP expression.

### 4.4. Aortic Morphometry

After fixation of the aortas in 4% formalin, the samples were dehydrated overnight and then treated with 70%, 90%, and 95% alcohol and then three baths of absolute alcohol (2 h at each concentration). The samples were then placed in equal parts alcohol and xylene overnight and cleared in xylene with changes every two hours. There were three changes performed and then embedded in paraffin.

The samples were cut transversely into 5-μm-thick sections in a microtome (RM 2255; Leica, Ltd. Cambridge, UK). Next, hematoxylin and eosin stains were performed to analyze atherosclerotic areas, cross-sectional area, media layer thickness, and lumen diameter. All images were obtained using an objective with a magnification of 10× under an optical microscope. The measurement was performed using Image J software 64 bit from National Institutes of Health (NIH).

### 4.5. Gelatinolytic Activity of MMP-2 in the Aorta by In Situ Zymography

We used the DQ gelatin substrate to quantify the activity of MMPs in vascular tissue. We first froze the aorta in a cryoprotection liquid (Tissue TeK, Sakura Finetek). Next, 5-µm-thick sections were formed in a cryostat (CM 1900; Leica, Imaging Systems Ltd. Cambridge, UK). The sections were incubated with DQ gelatin at a 100 µg/mL concentration for 1 h at 37° in a dark and humid chamber. The sections were then washed 3× for 5 min in PBS and then fixed with 2% formalin. The nuclei were next stained with DAPI. Images were acquired using a fluorescence microscope (Carl Zeiss Microscopy Ltd., Cambridge, UK). The gelatinolytic activity was quantified using Image J software measuring the fluorescence intensity in arbitrary units.

### 4.6. MMP-2 Expression in the Aorta by Immunofluorescence

To assess the expression of MMP-2 in vascular tissue, we next used sections of the aorta previously frozen in cryoprotection liquid (Tissue Tek, Sakura Finetek). The 5-µm-thick sections were prepared in a cryostat (CM 1900; Leica, Imaging Systems Ltd. Cambridge, UK). The material was first fixed in 2% paraformaldehyde for 15 min and then washed in PBS 3× (5 min per wash). Next, the tissue was blocked with 3% BSA containing 0.1% triton x-100 and 100 mM glycine for 30 min. Anti-MMP-2 primary antibody 1:500 (NB200-193, NovusBio, São Paulo, SP, Brazil) was incubated overnight at 4 °C. Subsequently, the sections were washed 5× in PBS with each wash for 5 min. Next, an anti-rabbit secondary antibody conjugated with Cy5 1:500 (GE Healthcare, 29038278) was used and incubated for 1 h at room temperature in a humid chamber. The nuclei were labeled with DAPI. We captured photos on a fluorescence microscope (Carl Zeiss Microscopy Ltd., Cambridge, UK). Red fluorescence was emitted by Cy5 fluorophore and was quantified using Image J software. The result was expressed as fluorescence intensity in arbitrary units.

### 4.7. ROS Determination In Situ

The aortas were frozen in cryoprotection liquid (Tissue Tek, Sakura Finetek, Torrance, CA, USA), and 5-µm-thick sections were formed in a cryostat (CM 1900; Leica, Imaging Systems Ltd. Cambridge, UK). The sections were incubated with dihydroethidium solution (DHE, 10 µmol/L, 30 min, at 37 °C) diluted in PBS (pH: 7.4). After incubation, the sections were washed 3× for 5 min in PBS (pH: 7.4) and photographed under a fluorescence microscope (Carl Zeiss Microscopy Ltd., Cambridge, UK) with a 40× objective. The ROS contents were expressed in arbitrary units of fluorescence intensity of the DHE probe.

### 4.8. Statistical Analysis

The results were expressed as the mean ± standard error (SEM) and analyzed with Graph Pad Prism ®️ 6.0 software (Graph Pad, San Diego, CA, USA). Data normality was assessed with the Shapiro–Wilk test, after a *t*-Student test was used. Pearson’s correlation test was performed. Values of *p* < 0.05 were considered statistically significant.

## 5. Conclusions

We conclude that doxycycline had an atherogenic effect in *ApoE*^-⁄-^/OVX animals, reducing lesions associated with a decrease in the MMP activity, MMP-2 expression, and ROS. There were no lipid-lowering effects. Our research corroborates previous findings. However, more studies are needed to establish adequate conditions for the clinical use of doxycycline.

## Figures and Tables

**Figure 1 ijms-23-02532-f001:**
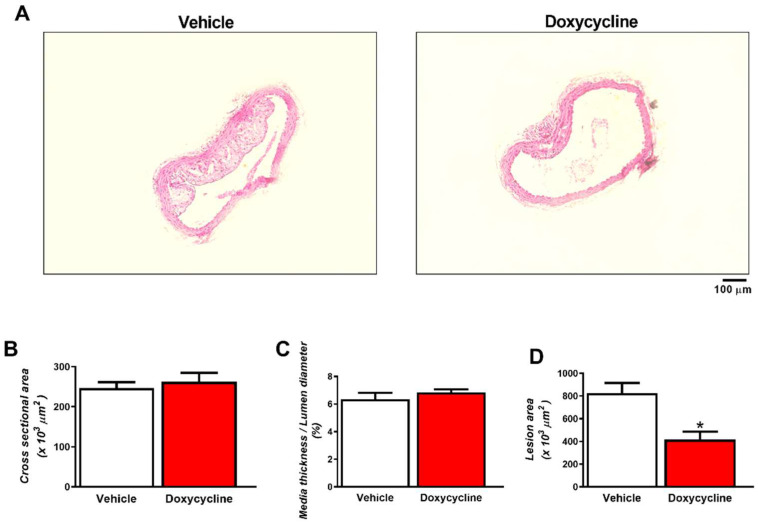
Doxycycline decreases atherosclerotic lesions in the thoracic aorta of *ApoE*^-⁄-^/OVX mice. (**A**) Representative white-light microscope photomicrographs of *ApoE*^-⁄-^/OVX vehicle and doxycycline-treated mice aortic sections stained with hematoxylin and eosin. Treatment with doxycycline decreased areas of atherosclerotic lesions in the aorta of *ApoE*^-⁄-^/OVX. (**B**) Quantification of the cross-sectional area of the aorta. (**C**) Quantification of the middle layer thickness/lumen diameter ratio. (**D**) Quantification of areas of atherosclerotic lesions. Values are expressed as mean ± SEM; * *p* < 0.05 vs. vehicle, *t*-Student test (*n* = 6 group). Scale bar: 100 µm.

**Figure 2 ijms-23-02532-f002:**
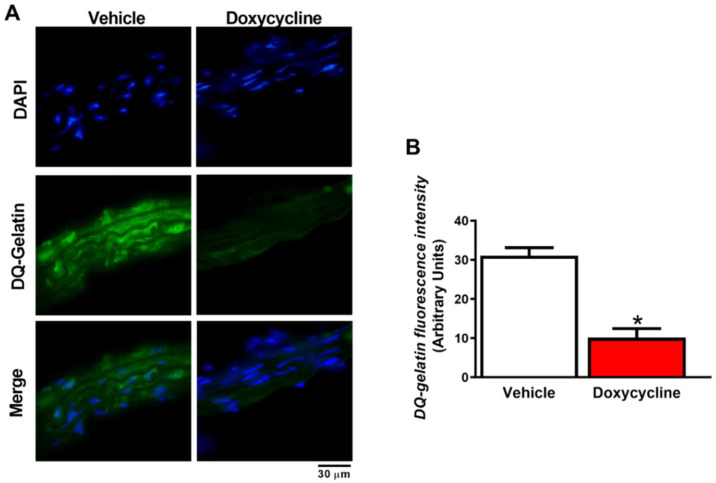
Doxycycline decreases MMP activity in the aorta of *ApoE*^-⁄-^/OVX mice. (**A**) Representative fluorescence microscope photomicrographs of mice aortic sections of *ApoE*^-⁄-^/OVX vehicle and doxycycline, incubated with DAPI and DQ-gelatin. DAPI’s blue fluorescence indicates nuclei, and green fluorescence represents MMP activity (DQ-gelatin), followed by the merge of the images showing the cell and vascular gelatinolytic MMP activity. (**B**) The gelatinolytic activity was quantified by measuring the intensity of green fluorescence. Values are expressed as mean ± SEM. * *p* < 0.05 vs. vehicle, *t*-Student test (*n* = 6 group). Scale bar: 30 µm.

**Figure 3 ijms-23-02532-f003:**
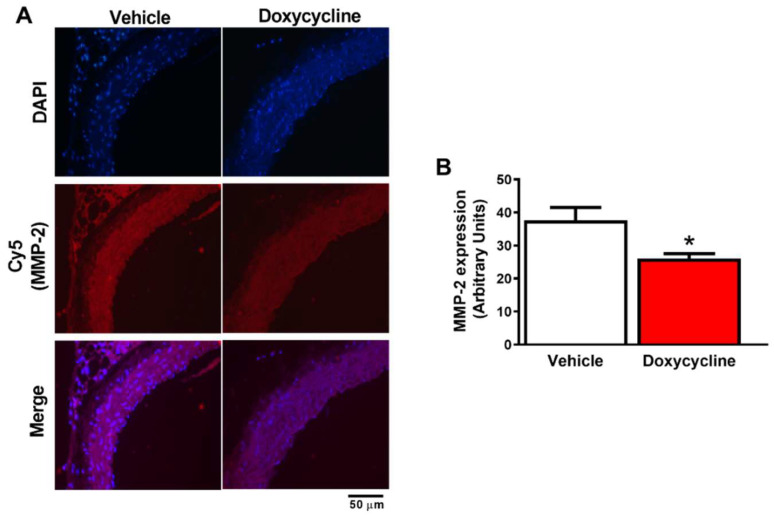
Doxycycline decreases MMP-2 expression in the aorta of *ApoE*^-⁄-^/OVX mice. (**A**) Representative fluorescence microscope photomicrographs of *ApoE*^-⁄-^/OVX vehicle and doxycycline mice aortic sections, incubated with DAPI and anti-MMP-2 antibody (Cy5 used as a secondary antibody). DAPI blue fluorescence indicates nuclei, and red fluorescence indicates MMP-2 labeling in vascular tissue, followed by the merge of the images showing the cell and MMP-2 staining. (**B**) Quantification of Cy5 fluorescence intensity (indicating MMP-2 expression). Values are expressed as mean ± SEM. * *p* < 0.05 vs. vehicle, *t*-Student test (*n* = 6 group). Scale bar: 50 µm.

**Figure 4 ijms-23-02532-f004:**
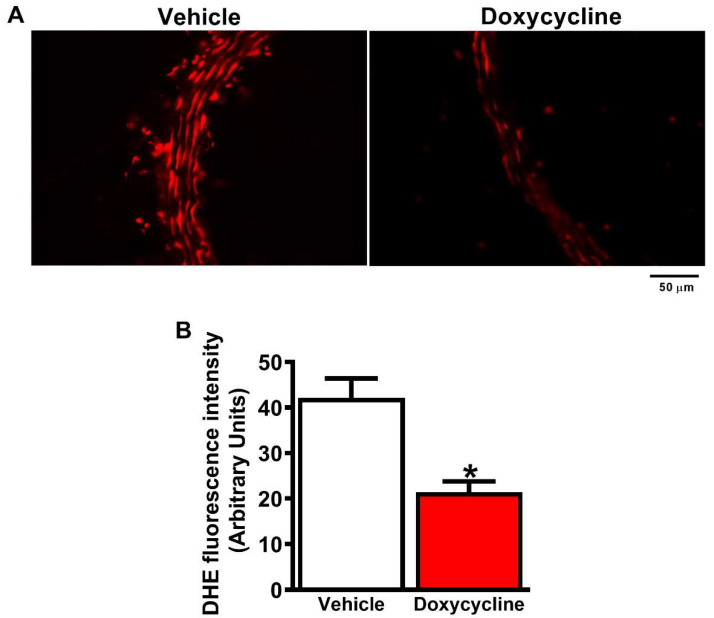
Doxycycline decreases ROS levels in the aorta of *ApoE*^-⁄-^/OVX mice. (**A**) Representative fluorescence microscope photomicrographs of *ApoE*^-⁄-^/OVX vehicle and doxycycline mice aortic sections, incubated with the DHE probe. Red fluorescence reflects ROS production, decreased in the *ApoE*^-⁄-^/OVX doxycycline group. (**B**) Quantification of DHE fluorescence intensity that reflects ROS levels. Values are expressed as mean ± SEM. * *p* < 0.05 vs. vehicle, *t*-Student test (*n* = 6 group). Scale bar: 50 µm.

**Figure 5 ijms-23-02532-f005:**
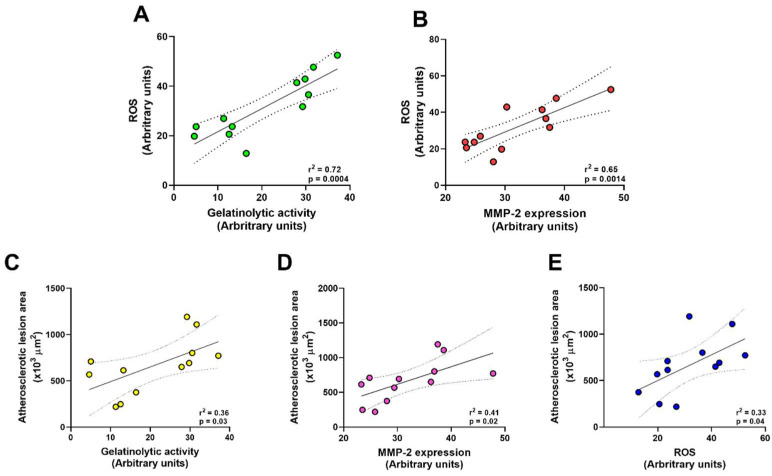
Doxycycline has an antiatherogenic effect correlated with decreased gelatinolytic activity, MMP-2 expression, and ROS levels in aortas of *ApoE*^-⁄-^/OVX mice. (**A**) Scatter plot showing the correlation between increased gelatinolytic activity and ROS levels. (**B**) Scatter plot showing the positive correlation between increased MMP-2 expression and ROS levels. (**C**) Scatter plot showing the positive correlation between the increase in atherosclerotic lesions associated with the rise in gelatinolytic activity. (**D**) Scatter plot showing the positive correlation between the increase in atherosclerotic lesions related to the increase in MMP-2 expression. (**E**) A scatter plot shows the positive correlation between an increase in atherosclerotic lesions associated with increased ROS levels. Correlation analyses were performed using the data reported in Figure 1, Figure 2, Figure 3 and Figure 4. Pearson’s correlation test was performed. The solid line represents the linear regression. Regression coefficient (r) and statistical significance (*p*) are reported for each test, *p* values < 0.05 were considered significant.

**Figure 6 ijms-23-02532-f006:**
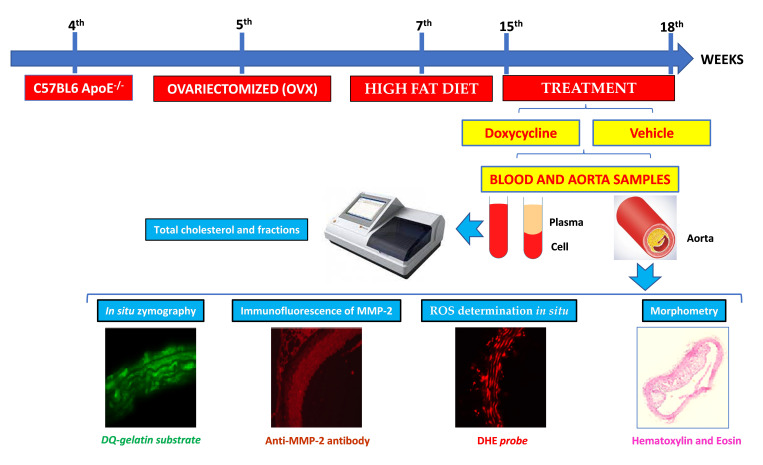
The workflow of the experimental design of the study. This study used four-week-old female C57Bl6 Knockout *ApoE*^-⁄-^ mice. In the 5th week of life, the animals underwent ovariectomy to induce experimental menopause. For induction of atherosclerosis, the hypercholesterolemic diet was instituted from the 7th week of life until the end of the experimental protocol. Treatments with vehicle (water) and doxycycline 30 mg/kg were administered by gavage once a day from the 15th week to the 18th week of life (28 days). After treatment, the animals were euthanized, and thoracic aorta and blood were collected. Aorta was used for morphometric analysis (using hematoxylin and eosin sections), gelatinolytic activity (using DQ-gelatin substrate), MMP-2 expression (using anti-MMP-2 antibody), and ROS (using DHE probe). Plasma samples were obtained for measurement of total cholesterol and fractions.

**Table 1 ijms-23-02532-t001:** Plasma values of total cholesterol, LDL, and HDL from *ApoE*^-⁄-^/OVX vehicle and doxycycline groups.

	*ApoE*^-⁄-^/OVX Vehicle	*ApoE*^-⁄-^/OVX Doxycycline
Cholesterol total (mg/dL)	799 ± 48	949 ± 47
Cholesterol LDL (mg/dL)	493 ± 54	411 ± 48
Cholesterol HDL (mg/dL)	5.9 ± 0.3	6.0 ± 0.5

Values were expressed as mean ± SEM. *p* > 0.05, *t*-Student test (*n* = 6 group).

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
