# Peer review of "Doxycycline Decreases Atherosclerotic Lesions in the Aorta of ApoE-⁄- and Ovariectomized Mice with Correlation to Reduced MMP-2 Activity"

_ijms, 2022, doi:10.3390/ijms23052532_

Round 1

Reviewer 1 Report

Comments:

  1. Could the authors please clarify how the numbers of animals for experiments were determined i.e. how was their study powered?
  2. Could the authors please justify the use of parametric statistical analyses? 

Reviewer 2 Report

This is the novel and first study that treatment with doxycycline in ApoE-/- / OVX animals reduced atherosclerotic lesions associated with MMP-2 activity and reduced ROS, such treatment is not associated with improved lipid profiles. Albeit, this review highlight the recent advances in aortic remodeling. I still have some minor suggestions.

  1. It would be much better if the authors can provide some Workflow or Scheme for this manuscript, I suggest that they can take a look at the recent paper in MDPI (PMID: 32992497, PMID: 34834441, PMID: 34679479).
  1. All figures are highly professional, and the authors should guide the readers to the meaning of the images appropriately; otherwise, it is likely to cause misunderstandings. Therefore, I suggest that the author consider revising these figure legends again.
  2. Gelatin zymography is the common method for examining matrix metalloproteinase-2 (MMP-2) in cells and media samples. It would be very interesting if the author can also discuss these information in discussion (PMID: 33921148, PMID: 34944720, PMID: 34885656).

Reviewer 3 Report

The results briefly exposed by the Authors on the action of doxycycline toward the reduction in atherosclerotic lesions in ApoE-/-/ OVX animals is new and deserved publication.

However, the style has to be revised. If the direct style used for the results could be accepted. The style used for the discussion is less usual. In its present form, it is difficult to have a clear comparison of the results of the authors against those from the litterature data.

Add a conclusion paragraph after Materials and Methods as suggested by the Instructions to Authors could be of interest as well.
